# DT-PICS: An Efficient and Cost-Effective SNP Selection Method for the Germplasm Identification of Arabidopsis

**DOI:** 10.3390/ijms24108742

**Published:** 2023-05-14

**Authors:** Liwen Xiong, Zirong Li, Weihua Li, Lanzhi Li

**Affiliations:** Hunan Engineering & Technology Research Center for Agricultural Big Data Analysis & Decision-Making, College of Plant Protection, Hunan Agricultural University, Changsha 410128, China; xlw_1112@163.com (L.X.); lizirong903@163.com (Z.L.);

**Keywords:** germplasm identification, Arabidopsis thaliana, SNP, DNA fingerprinting

## Abstract

Germplasm identification is essential for plant breeding and conservation. In this study, we developed a new method, DT-PICS, for efficient and cost-effective SNP selection in germplasm identification. The method, based on the decision tree concept, could efficiently select the most informative SNPs for germplasm identification by recursively partitioning the dataset based on their overall high PIC values, instead of considering individual SNP features. This method reduces redundancy in SNP selection and enhances the efficiency and automation of the selection process. DT-PICS demonstrated significant advantages in both the training and testing datasets and exhibited good performance on independent prediction, which validates its effectiveness. Thirteen simplified SNP sets were extracted from 749,636 SNPs in 1135 Arabidopsis varieties resequencing datasets, including a total of 769 DT-PICS SNPs, with an average of 59 SNPs per set. Each simplified SNP set could distinguish between the 1135 Arabidopsis varieties. Simulations demonstrated that using a combination of two simplified SNP sets for identification can effectively increase the fault tolerance in independent validation. In the testing dataset, two potentially mislabeled varieties (ICE169 and Star-8) were identified. For 68 same-named varieties, the identification process achieved 94.97% accuracy and only 30 shared markers on average; for 12 different-named varieties, the germplasm to be tested could be effectively distinguished from 1,134 other varieties while grouping extremely similar varieties (Col-0) together, reflecting their actual genetic relatedness. The results suggest that the DT-PICS provides an efficient and accurate approach to SNP selection in germplasm identification and management, offering strong support for future plant breeding and conservation efforts.

## 1. Introduction

*Arabidopsis thaliana*, as a model plant, has great value in modern genetics [1]. Before the formal start of research, confirmation of germplasm identity is an essential step [2]. Misidentified, the germplasm resource will most likely result in a significant waste of human resources and time, limiting the potential for genetic analysis [3]. Moreover, germplasm confusion is also common in seed stock [4]. Using phenotypic traits is the simplest and fastest method of germplasm identification. However, it is susceptible to external environmental influences [5]. Unlike biochemical and morphological markers, molecular markers are environmentally stable, providing significantly high genetic polymorphism while allowing for analysis at any developmental stage [6]. Molecular markers are now widely used for variety identification in plants such as rice [7], Arabidopsis [2], cucumber [8], soybean [9], and so on.

Given the stability and effectiveness of germplasm identification methods, the International Union for the Protection of New Varieties of Plants (UPOV) has identified DNA molecular markers as SSR and SNP marker [10,11]. However, SNP has a higher potential for application than SSR based on its advantages. In general, SNPs are widely distributed and highly abundant in the genome, present high genetic stability and good repeatability, and allow high-throughput automated analysis [8,12]. With the development of second-generation sequencing technology, SNP has rapidly been applied in various crops for genetic variation [13] and diversity analysis [14], which is regarded as one of the most important and promising markers [15].

Using SNPs for the germplasm identification of Arabidopsis has been applied to some extent. The Arabidopsis germplasm identification tools AraGeno and SNPmatch, developed by Rahul Pisupati et al. and based on the 1001 Genomes Project, can be used directly online to identify germplasm to be tested by comparing its whole-genome SNPs with a reference database of 10.7 million SNPs. However, it is based on whole genome alignments. Additionally, the identification accuracy is highly dependent on the number of markers in the germplasm to be tested, which decreases as the number of markers decreases. It may fail to identify germplasm when the number of markers is less than 10,000 [2]. Relative to Pisupati et al.’s method, Matthieu Simon et al. used only 341 SNPs to distinguish 1311 Arabidopsis seeds corresponding to 598 varieties and developed a tool called ANATool. For marker selection, they first integrated markers previously developed by other researchers, then manually screened markers evenly distributed on chromosomes with intermediate allele frequencies (the more intermediate the frequency is, the more discriminating the marker is) [16]. They further screened the markers with locus detection rates and GenTrain scores to obtain a set of 341 SNPs. However, the marker screening process was not automated and was somewhat dependent on the validity of markers selected by other researchers [3]. Similar to intermediate allele frequencies, the Polymorphism Information Content (PIC) values were also considered in many other studies [8], since intermediate allele frequencies tend to be more highly associated with high PIC values [16]. The variety distinction pursuit involves low labor costs and high efficiency, which means distinguishing all varieties with the minimum number of SNPs [15]. Ideally, approximately 1000 Arabidopsis varieties can be distinguished with a set of 10 binary SNPs. Many previous studies only selected individual SNPs with high PIC values, without considering their overall performance, which ignores the redundant information between selected markers. Thus, there is still room for the further streamlining of existing selected markers.

To achieve a strong overall distinction, it is recommended that markers be complementary to each other. The decision tree method (DT) is one of the well-known methods for data classification. The most distinctive markers can be quickly selected by DT for recursive partitioning and the redundancy between markers can be reduced; thus, the core features can be quickly filtered out for classification [17]. Although germplasm identification is not a typical classification problem, large, complicated datasets can be efficiently dealt with using the DT method without imposing a complicated parametric structure. The concept of the DT method could be used in the germplasm identification process [18] since the method can facilitate faster screening and enable automation of the germplasm identification process.

In this study, we developed a DT-PICS (Decision Tree-PIC Selection) method for Arabidopsis germplasm identification. At first, for 1135 Arabidopsis varieties, the simplified SNP set with an overall high PIC value was screened out with DT-PICS. Then, multiple simplified SNP sets were combined to construct the fingerprint map to enhance the fault tolerance in independent identification. We then compiled all the code into R scripts for automatic analysis. This method has good portability and can also be used for the fingerprint construction of other plants.

## 2. Results

### 2.1. Characteristics of SNPs in Training Data

The distribution of PIC values for the 736, 195 cSNPs in Train_Data (736, 195 × 1135) shown in Figure 1A are relatively uniform. Using the DT-PICS method, we obtained 13 simplified SNP sets that contained a minimum of 52 SNPs that could distinguish the 1135 Arabidopsis germplasms, with an average of 59 SNPs per set (Appendix A). These 13 sets contained a total of 769 DT-PICS SNPs that were evenly distributed across the chromosomes. Among these 769 SNPs, 170 SNPs (22.11%) had high PIC values (PIC value > 0.45), while 160 SNPs (20.81%) had low PIC values (PIC value < 0.2). (Figure 1B).

### 2.2. Fingerprint of 1135 Arabidopsis Varieties Established by DT-PICS

Figure 2A shows that when using a single SNP set to validate the distinguishing accuracy of marker selection methods at five SNP modification levels, the DT-PICS had a higher distinguishing accuracy (92.15%) than High PIC-value Selection (HPS, 82.86%) and Random Selection (RS, 77.77%) on average, indicating the superior identification ability of DT-PICS. Additionally, the distinguishing accuracy of the 13 simplified SNP sets at different SNP modification levels is shown in Figure 2B. We observed a decrease in distinguishing accuracy as the proportion of SNP modifications increased. Among the five SNP combinations, the 13 simplified SNP sets exhibited similar high distinguishing accuracy, with the 11th SNP set (containing 59 SNPs) achieving the highest average distinguishing accuracy (93.54%) and the best fault tolerance (Table 1).

Table 1 shows a comparison of identification simulation analyses of the 11th SNP set. The order of average distinguishing accuracy of the three methods was DT-PICS (93.54%) > HPS (82.92%) > RS (78.32%). The distinguishing accuracy using the 11th SNP set was 96.74%, 92.78%, and 84.64% at 5%, 10%, and 15% SNP modification levels, respectively. The mean distinguishing accuracy of the SNP set was 91.39% with a standard deviation of 0.062.

When combining multiple simplified SNP sets, a significant increase in distinguishing accuracy was observed according to SNP modification levels (Figure 2C), indicating that combined multiple simplified SNP sets could enhance the markers’ fault tolerance and stability in practical application effectively. Additionally, the degree of inconsistency in SNP changes may vary depending on the sequencing platform and sample used. SNP changes were not significant with advancements in sequencing technology [8]; thus, using a level of 5% SNP modification may better reflect practical scenarios. Although the distinguishing accuracy increased with the use of more simplified SNP sets, the accuracy did not increase much from two sets (99.05%) to five sets (99.98%) at a 5% SNP modification level (Figure 2C, Appendix A). Therefore, to save costs while ensuring high accuracy, two simplified SNP sets were chosen for joint identification in this study.

We performed simulation experiments by combining the 13 simplified SNP sets in all possible pairwise combinations for joint identification, resulting in 78 combinations, and the DT-PICS method showed the best performance in both the original and modified SNP datasets (Appendix A). Among these combinations, the combination of the seventh and eighth simplified SNP sets containing 109 SNPs showed the highest distinguishing accuracy. The distinguishing accuracy of this combination was 99.44%, 97.94%, and 95.64% at 5%, 10%, and 15% SNP modification levels, respectively. The distinguishing accuracy of HPS and RS methods were similar, both being less than 90%. With several DT-PICS SNPs instead of a portion of hPIC SNPs (PIC value > 0.45), the distinguishing accuracy significantly increased (Table 2). This further highlights the superiority of the DT-PICS method.

### 2.3. Fingerprint of the Identification Power of Same-Named and Different-Named Varieties in Test Datasets

In independent predictions, for 68 same-named varieties, the variety with the highest similarity was recorded as the identification result. The DT-PICS method showed the highest identification accuracy, with an average of 94.97%, and only 30 shared markers between training and test datasets on average (Figure 3A, Appendix A). In contrast, the HPS and RS methods exhibited almost zero identification accuracy. Comparing the similarities of the 68 same-named varieties in the training and test datasets showed that the similarity with DT-PICS SNPs (80.95~100%) reached a level comparable to that with the whole genome SNPs (83.76~100%), indicating that the SNPs selected by the DT-PICS method could effectively represent the genetic diversity of these germplasms (Figure 3C). Two varieties, ICE169 and Star-8, were separately incorrectly identified as ICE173 and Uk-3 with DT-PICS SNPs. A comparison of the whole genetic similarity between these varieties demonstrated that ICE169 and Star-8 in Test_Data separately had a genetic similarity of 98.50% and 99.34% with their same-named varieties in Train_Data and of 99.08% and 99.35% with the incorrectly identified varieties in Train_Data. The identified varieties in Test_Data had a higher genetic similarity with other varieties rather than their corresponding same-named varieties, possibly indicating sample confusion or mislabeling, which requires further investigation.

Furthermore, all 12 different-named varieties were able to be distinguished from each other using DT-PICS SNPs. However, when each of these varieties was individually added to the Train_Data, they could be distinguished from the other 1134 varieties, but were always classified as the same variety as col-0. Further investigation into their genetic similarity with Col-0 at the whole-genome level revealed an average similarity of 99.99%. It is suggested that these 12 different-named varieties could be regarded as derivatives of Col-0 rather than independent new varieties. The DT-PICS method can effectively distinguish between the germplasm to be tested and other varieties while grouping extremely similar varieties together, reflecting their actual genetic relatedness. In contrast, the HPS and RS methods may sometimes classify a single germplasm into multiple different varieties, which results in increased uncertainty in the classification process. These results further demonstrate the superiority of DT-PICS for germplasm identification.

To demonstrate the flexibility of DT-PICS SNP selection, a gradient approach was used by randomly selecting a certain number of markers from all 769 DT-PICS SNPs to distinguish the 80 varieties in Test_Data. The distinguishing accuracy gradually increased with increases in the number of selected SNPs, reaching stability at 70 SNPs. Additionally, identification accuracy did not show any significant difference with same-size randomly selected DT-PIC SNPs, indicating that marker selection could be based on practical needs in practical applications, rather than being limited to specific marker sets (Figure 3B). These findings can help optimize SNP selection strategies to better meet practical application requirements and highlight the practicality of the DT-PICS method.

### 2.4. Generation of QR Codes

QR codes for 1135 Arabidopsis varieties were created. Each code contained the variety’s name and its DT-PICS SNP from two simplified SNP sets (Figure 4).

## 3. Discussion

Germplasm identification is crucial for managing genetic diversity in germplasm resources [5]. However, with the rapid growth of germplasm resources [19], germplasm confusion is common in stock centers [2,3,6]. Confirmation of germplasm identity before formal experimentation is essential, as errors due to mixing or mislabeling can occur over time [14]. While the variety discrimination pursuit involves low labor costs and high efficiency, there is still room for the further streamlining of existing markers. Our research aims to make the process of germplasm identification more efficient and cost-effective.

In this study, with the strategy of the DT method concept for “divide and conquer” and “greedy” based on PIC values, a DT-PICS germplasm identification method based on the SNPs of the whole genomes of 1135 Arabidopsis varieties was developed. In contrast to most existing marker selection methods that focus solely on individual SNP’s high PIC values [20,21,22], our study considered overall high PIC values (PIC_sum_) that take into account both the overall performance of the SNP dataset and the unique characteristics of each marker. This approach allowed us to avoid selecting redundant SNPs with high PIC values that do not provide additional discriminatory power and to identify low-PIC-value SNPs that contain novel information and are valuable for germplasm identification. Additionally, the DT method could efficiently select the most informative SNPs for germplasm identification by recursively partitioning the dataset based on their overall high PIC values, which can reduce the cost and time required for genotyping large numbers of varieties [17,18].

Remarkably, when markers are selected using DT-PICS, the greedy algorithm may result in an uneven distribution of markers across sets, altering the population structure within the sample. This can increase the genetic distance between initially similar varieties and decrease it between more distinct varieties, rendering the set of markers less suitable for assessing variety similarity. Therefore, DT-PICS SNP is only applicable for germplasm identification but not suitable for comparing similarities between varieties or cluster analyses. Thus, DT-PICS SNP has a better distinguishing ability for similar varieties.

Using the DT-PICS method, 13 simplified SNP sets containing a total of 769 DT-PICS SNPs were generated, with an average of 59 SNPs per set, which were capable of distinguishing all 1135 Arabidopsis germplasms. Our results show that the DT-PICS method performed well in both Train_Data and Test_Data, indicating its superior identification ability for both same-named and different-named germplasms compared to the HPS and RS marker selection methods. Additionally, we observed a significant increase in distinguishing accuracy when using a portion of DT-PICS SNPs instead of hPIC SNPs. The gradient experiment demonstrated that a randomly selected set of DT-PICS SNPs (approximately 70 SNPs) achieved high distinguishing accuracy, which further highlights the flexibility and practicality of the DT-PICS method. Furthermore, we identified two potentially mislabeled varieties (ICE169 and Star-8), suggesting the importance of further investigation to ensure the accurate identification of germplasms.

Although the DT-PICS method has demonstrated its scientific validity and practicality, there are still some aspects that can be further improved. On the one hand, in Test_Data for the same-named varieties, the one with the highest similarity was selected as the identification result. However, for some cases where the same-named varieties’ similarity was not very high (Figure 3C), such as ICE107, the similarity of DT-PICS (80.95%) was significantly lower than that of the whole genome (98.99%); thus, further checks of the germplasm using other methods such as phenotype analysis may be necessary. On the other hand, although the 7th and 8th simplified SNP sets were chosen to construct the QR codes in this study, the flexibility in marker selection allowed for the use of alternative marker combinations. However, due to the lack of established standards for marker selection, the marker combinations used in this study may require further validation and optimization to improve the accuracy and reliability of germplasm identification. Therefore, future research can explore more standardized and stable marker combinations to better achieve germplasm identification and management.

## 4. Materials and Methods

### 4.1. Genotype Dataset of Materials

There were two Arabidopsis SNP genotype datasets involved in this study: one training dataset and one test dataset (Figure 5). The training dataset was downloaded from the 1001 Genomes Project (https://1001genomes.org, accessed on 4 April 2022) [23,24], including more than 119 million SNPs of 1135 Arabidopsis germplasms worldwide. The SNPs that passed quality control were maintained by removing SNPs with >20% missing calls and MAF < 5% and using a two-step linkage disequilibrium pruning procedure with PLINK (version 1.9) [25]. Then, SNP genotype imputation was performed with beagle software (version 5.1) [26], and 1,158,135 SNPs remained.

SNPs include coding-region SNPs (cSNPs), intergenic SNPs (iSNPs), and perigenic SNPs (pSNPs). The variation rate of cSNPs within exons is only 20% of that of other surrounding sequences. Although the number of them is relatively small, they are important in the study of biological breeding and genetic diseases. Therefore, in this study, we screened cSNPs located in the coding region of genes for further Arabidopsis germplasm identification. Arabidopsis gene annotation information was obtained from the genome database of NCBI GenBank (http://www.ncbi.nlm.nih.gov/, accessed on 14 April 2022). Then, 749,636 cSNPs located in the gene’s coding region were extracted and used for following Arabidopsis fingerprint mapping modeling. SNPs that were genotypically identical in all samples were removed. The training dataset contained 1135 varieties with 736, 195 cSNPs, denoted as Train_Data (736, 195 × 1135).

The test dataset contained 80 Arabidopsis varieties downloaded from the MPICao2010 of 1001 Genomes Project (https://1001genomes.org, accessed on 5 May 2022), including 68 same-named and 12 different-named varieties as those in Train_Data [27]. Then, quality control and cSNPs extraction were conducted, as with the training dataset. The test dataset contained 80 varieties with 119,411 cSNPs, denoted as Test_Data (119, 411 × 80). For same-named and different-named varieties, two accuracy metrics, Identification Accuracy and Distinguishing Accuracy, were used to evaluate the identification ability of the method.
$$Identification\;Accuracy=\frac{Number\;of\;varieties\;identified\;correctly}{Total\;number\;of\;tested\;varieties}\times 100\%$$
$$Distinguishing\;Accuracy=\frac{Number\;of\;varieties\;distinguished}{Total\;number\;of\;tested\;varieties}\times 100\%$$

### 4.2. Marker Polymorphism Analysis

The PIC of each cSNP in Train_Data was calculated with the software R language. The calculation formula is as follows:(1)$$PIC=1-\sum f_{i}^{2}$$
where f is the genotype frequency of the ith SNP [28].

### 4.3. Selecting SNPs from the Training Dataset to Construct a Fingerprint Map

The flowchart of the DT-PICS method is shown in Figure 6. To explain the concept of the DT-PICS method clearly, the dataset with eight varieties (V1~V8) and seven SNPs was taken as an example, shown in a schematic diagram (Figure 7).

Stage 1: preliminary screening of SNP markers. First, the SNPs in Train_Data were sorted with the PIC values in descending order. One SNP was randomly selected from the top 10% of SNPs. According to the genotype of the SNP, Train_Data was split into two datasets, split_1 and split_2, and two leaf nodes were obtained. As shown in Figure 7, the SNP1 was selected and the original dataset was split into two datasets.

The PIC values of the remaining SNPs were recalculated in each split dataset (split_1, split_2…split_l), and the PIC values of all SNPs in each split dataset were summed to obtain ${PIC}_{sum}$. ${PIC}_{sum}=\sum_{j=1}^{l}\sum_{i=1}^{2}(1-P_{ij}^{2})$, where l is the number of datasets to be split and P_ij_ is the genotype frequency of the ith marker (*i* = 1~*m*, m is the total number of markers) of the *j*th split dataset (*j* = 1~*l*). The remaining SNPs were sorted with the values of ${PIC}_{sum}$ in descending order. Then, another SNP was randomly selected from the top 10% of the remaining SNPs. The previously split datasets were further split into new sub-datasets according to the genotype of the new selected SNP.

The above steps were repeated until the number of leaf nodes and the number of split datasets equaled the total number of samples *n*. As shown in Figure 7, SNP1 and SNP3 were separately selected in the first and second rounds, and the datasets were split into four leaves. After four rounds, the raw dataset was split into eight leaves according to the genotypes of the four selected SNPs (Figure 7), which meant that the set of these SNPs (M) could distinguish all varieties.

Stage 2: Redundant SNP deletion. One SNP in the M set was randomly shielded at a time. If the remaining selected SNPs could still distinguish all varieties, this indicated that the shielded SNP was redundant and should be filtered out. Otherwise, the SNP remained in the M set. We repeated this step several times until all the markers in the M set had been checked. As shown in Figure 7, SNP 2 was a redundant SNP. The genotype combinations of SNP1, SNP3, and SNP4 could still distinguish all eight varieties, forming a simplified SNP set.

### 4.4. Variety Identification in the Training Set Using the DT-PICS Method

To evaluate the effectiveness of germplasm identification with the above simplified SNP sets, markers selected from three methods, DT-PICS, HPS (High PIC selection), and RS (Random Selection), were separately performed for germplasm identification in simulation analyses. The HPS method selected SNPs with PIC values > 0.45 (hPIC) in Train_data as high PIC SNPs; the RS method selected SNPs randomly from Train_Data for identification. Five experimental levels, with all DT SNPs, 2/3 DT SNPs + 1/3 hPIC SNPs, 1/3 DT SNPs + 2/3 hPIC SNPs, all hPIC SNPs, and all RS SNPs, were set up with five repeats at each level. The number of markers at each level equaled the average value of all simplified SNP sets selected by DT-PICS mentioned above. To compare the distinguishing accuracy of the three methods, we separately modified 5%, 10%, and 15% SNPs of the germplasm to simulate the inconsistency of SNP genotypes among different samples of the same germplasm that can be caused by mutations, deletions, differences in sequencing platforms, and other factors in practical applications.

For some selected markers that may not have been detected in the tested germplasm during the SNP calling in different sequencing platforms, the accuracy of independent prediction was reduced due to insufficient core markers. As a result, multiple simplified SNP sets should be combined for independent prediction to improve the stability and fault tolerance of the DT-PICS. The combination of multiple simplified sets should contain a relatively small size of markers and can well distinguish germplasms from each other in practical applications. To save identification costs as much as possible, it is necessary to determine the optimal number of simplified SNP sets. Similar to the analysis method using a single simplified SNP set mentioned above, 5%, 10%, or 15% SNP genotypes of tested germplasm were randomly modified, then combined with several simplified SNP sets to determine the optimal combination for independent germplasm identification.

### 4.5. Independent Testing of Variety Identification

Three methods (DT-PICS, RS, and HPS) were used to identify 68 same-named and 12 different-named varieties in Test_Data, as in Train_Data. The prediction accuracy was evaluated based on the identification accuracy and the distinguishing accuracy mentioned above.

### 4.6. Generation of QR Codes

QR codes were generated for the varieties used in this study using two combinations of simplified SNP sets that had shown the best performance in the training set. The codes were generated using an online tool available at www.barcode-generator.org (accessed on 13 March 2023) [11]. Each variety’s name and its DT-PICS SNP from two simplified SNP sets were used as input to generate corresponding QR codes. Once the code had been generated, it was scanned for the confirmation of information used for germplasm identification.

## 5. Conclusions

This study presents a novel DT-PICS method for quickly and accurately identifying varieties, which offers several advantages over existing methods. The approach is flexible and practical, with no need for manual selection and a low number of markers. Moreover, an Rscript (Appendix A) was compiled to make the marker screening process more manageable and applicable, enabling the automatic construction of fingerprint maps for the germplasm identification of other plants. SNP-based germplasm identification technology has great potential for identifying new and existing varieties, and our method can provide technical support for constructing relevant germplasm fingerprint maps. This research has significant implications for developing future marker screening methods in this field.

## Figures and Tables

**Figure 1 ijms-24-08742-f001:**
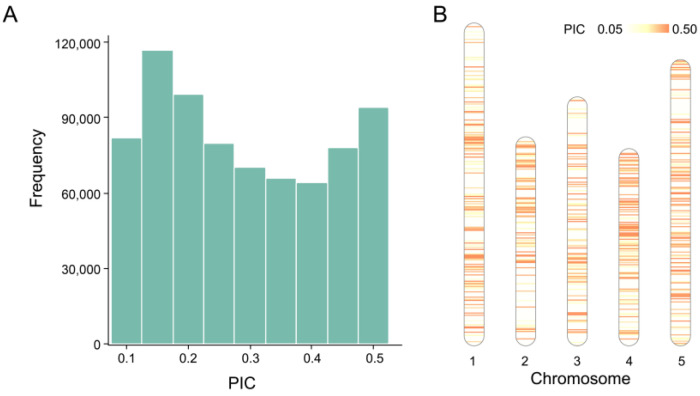
The PIC value of cSNPs in Train_Data and the DT-PICS-selected SNP distribution on chromosomes. (**A**) The histogram of PIC values of cSNPs in Train_Data; (**B**) The chromosome distribution of 769 SNPs screened out with DT-PICS.

**Figure 2 ijms-24-08742-f002:**
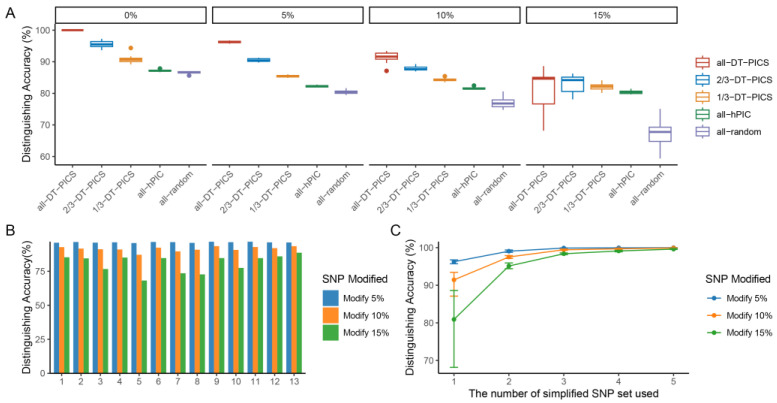
Comparative analysis of the distinguished accuracy of 13 simplified SNP sets. (**A**) The identification ability of three methods in training data by individual simplified SNP sets. (**B**) The identification ability of 13 simplified SNP sets. (**C**) Combined multiple simplified SNP sets for identification.

**Figure 3 ijms-24-08742-f003:**
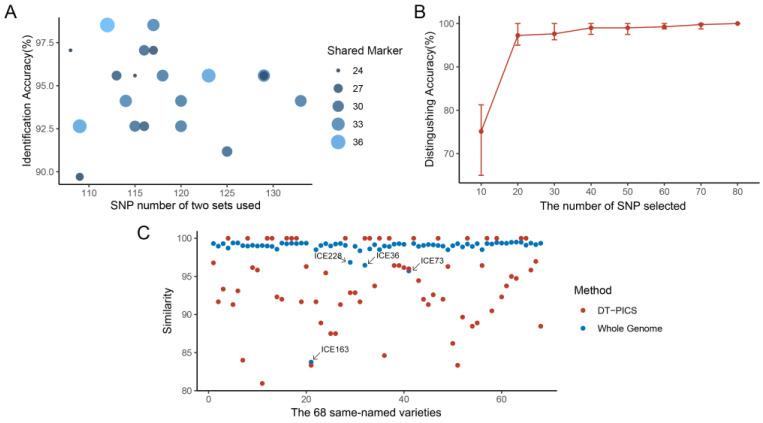
Independent validation results of Test_Data. (**A**) Identification accuracy of the 68 same-named varieties, with 20 repeats, plotted against the number of markers included in two sets of markers. The size of the dots indicates the number of shared markers used in identification. (**B**) Gradient experiment using a random selection of markers from the 769 SNPs to distinguish the varieties in Test_Data. (**C**) The similarity between the 68 same-named varieties using DT-PICS SNPs and whole-genome SNPs.

**Figure 4 ijms-24-08742-f004:**
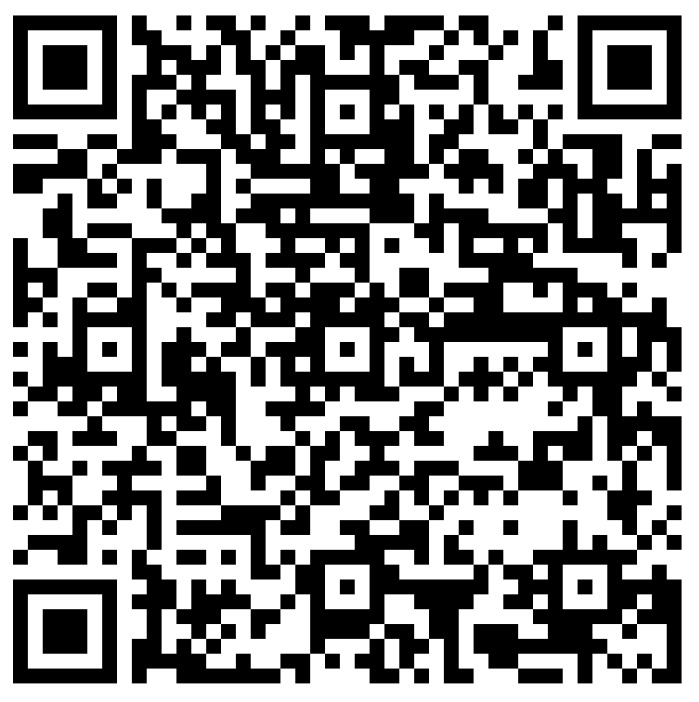
QR code of a representative variety of Arabidopsis used in the present study.

**Figure 5 ijms-24-08742-f005:**
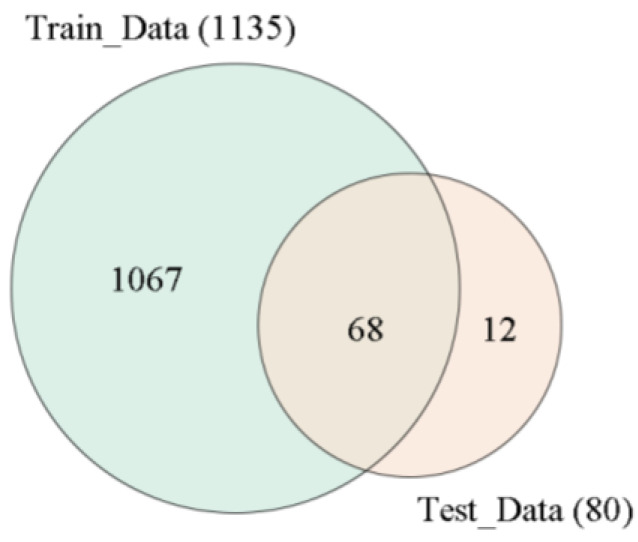
Venn diagram for the two datasets.

**Figure 6 ijms-24-08742-f006:**
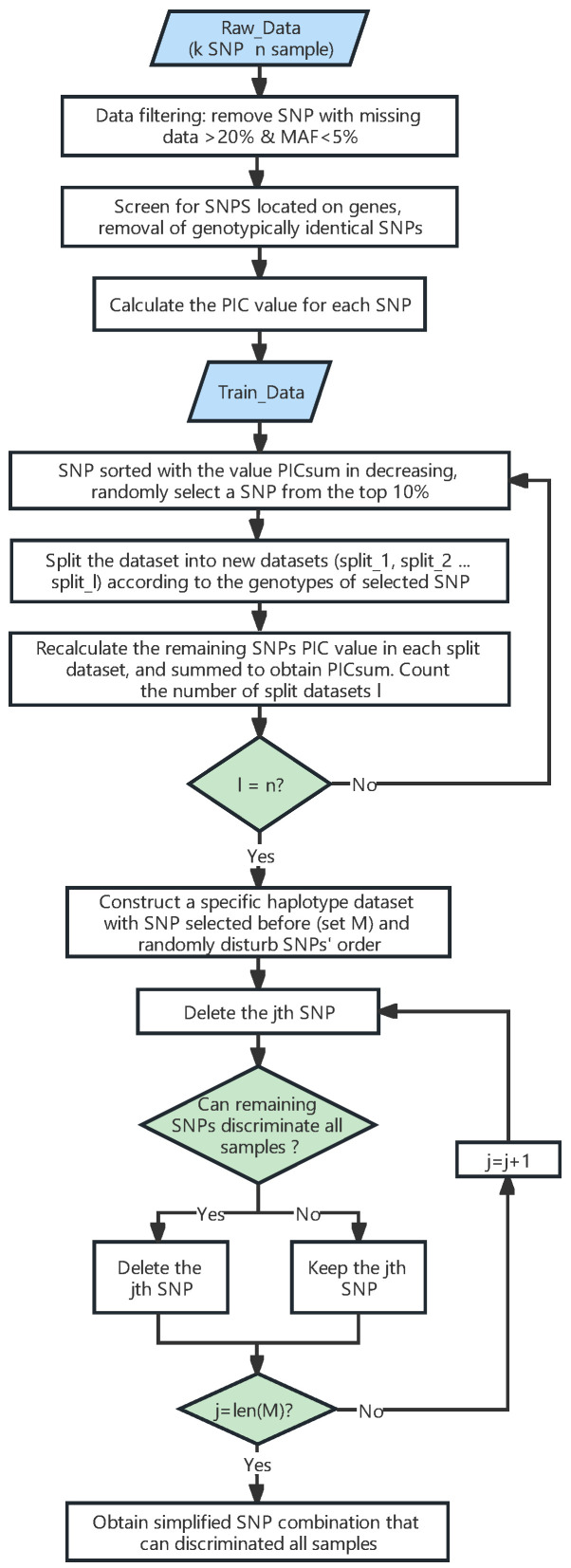
Flow chart of the Decision-Tree PIC-Selecting SNP method.

**Figure 7 ijms-24-08742-f007:**
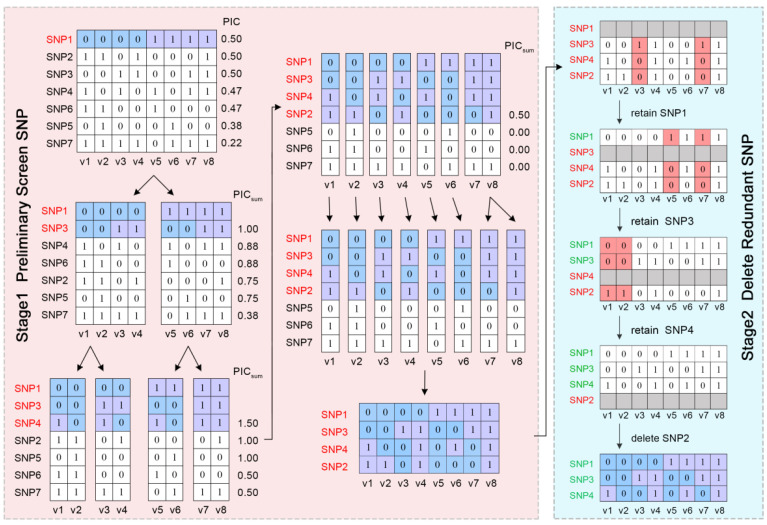
Schematic of the two stages of the Decision-Tree PIC-Selecting SNP workflow. The red part is the stage1 for preliminary screen SNP, the blue part is the stage2 for delete redundant SNP.

**Table 1 ijms-24-08742-t001:** Comparison of identification simulation analysis using the 11th simplified SNP set.

SNP Combination	Distinguishing Accuracy (%)				Mean Accuracy (%)
	Raw SNP	Modify 5% SNP	Modify 10% SNP	Modify 15% SNP	
59 DT-PICS SNPs	100	96.74	92.78	84.64	93.54
39 DT-PICS + 20 hPIC	94.19	91.28	88.65	85.21	89.83
20 DT-PICS + 39 hPIC	91.28	85.94	84.68	82.67	86.14
59 hPICS SNPs	87.14	82.32	81.71	80.51	82.92
59 random SNPs	86.70	80.42	77.90	68.26	78.32

**Table 2 ijms-24-08742-t002:** Comparison of identification simulation analysis using the 7th and 8th combination sets.

SNP Combination	Distinguishing Accuracy (%)				Mean Accuracy (%)
	Raw SNP	Modify 5% SNP	Modify 10% SNP	Modify 15% SNP	
109 DT-PIC SNPs	100	99.44	97.94	95.64	98.26
73 DT-PIC + 36 hPIC	99.56	97.28	94.82	92.56	96.06
36 DT-PIC + 73 hPIC	95.51	92.02	90.19	88.58	91.58
109 hPIC SNPs	87.93	83.86	83.57	83.14	84.63
109 random SNPs	87.58	83.10	82.57	81.59	83.71

## Data Availability

The raw data for this study can be found in the 1001 Genomes project database for GMI-MPI (accessed on 4 April 2022), MPICao2010 (accessed on 5 May 2022), and GCF_000001735.4_TAIR10.1 (accessed on 14 April 2022) on NCBI. The URL is https://www.ncbi.nlm.nih.gov/ and https://1001genomes.org/.

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
