# Peer review of "DT-PICS: An Efficient and Cost-Effective SNP Selection Method for the Germplasm Identification of Arabidopsis"

_ijms, 2023, doi:10.3390/ijms24108742_

Round 1
Reviewer 1 Report
The authors developed a new method for SNP selection in germplasm identification that reduces redundancy in SNP selection and enhances the efficiency and automation of the selection process. The method select the most informative SNPs for germplasm identification based on their overall high PIC values, instead of considering individual SNP features. The article is written clearly, well-structured and the references are recent and adequate. The developed method can be used in similar studies with other plants. I am the opinion that the work can be published.
Minor changes:
Lines 17 and 24 – standardize the style 1,135 and 1134.
Line 170 - erase the comma.
Lines 172 and 173 – standardize two decimal places.
Author Response
Point 1: Lines 17 and 24 – standardize the style 1,135 and 1134.
Response 1: The format here has been standardized to 1,135 and 1,134.
Point 2: Line 170 - erase the comma.
Response 2: That has been erase the comma.
Point 3: Lines 172 and 173 – standardize two decimal places.
Response 3: The format here has been standardized to two decimal places.
Reviewer 2 Report
Positive notes:
1. The subject of the manuscript is within the scope of the International Journal of Molecular Sciences;
2. The topic is relevant because it concerns the experimentation of efficient and cost-effective SNP selection method for germplasm identification of Arabidopsis;
3. The abstract meets the requirements of the journal and represents the main scientific achievements in the manuscript;
4. The literature review and cited scientific articles show the level of science up to the present moment and the main challenges in the researched field;
5. The research methods used are up-to-date and compatible with the experimentation of a novel method DT-PICS for fast and accurate identifying varieties, which offers several advantages over existing methods;
6. The results are well presented and illustrate the innovations achieved in the study;
7. An in-depth attempt was made to discuss the obtained results and compare them with what has been achieved in the relevant scientific field;
8. It is concluded that this study presents a novel method DT-PICS for fast and accurate identifying varieties, which offers several advantages over existing methods. According to the authors, the approach is flexible and practical, with no need for manual selection and a low number of markers.
Negative notes and recommendations to the authors:
1. The article has not yet been finalized according to the magazine's requirements. For example, rows: 32, 41, 51, 52, 72, 214, 224, 303, 309;
2. Please zoom in on Figures 1 and 3 for a better visualization of the results;
3. Since A. thaliana is widely used in the fields of plant science, genetics and evolution and has helped further our understanding of germination and aspects of plant growth that are important in commercial crops, I recommend future research in this area.
The quality of the English language is reasonably good. Minimal revision by an English-speaking editor is required.
Author Response
Point 1: The article has not yet been finalized according to the magazine's requirements. For example, rows: 32, 41, 51, 52, 72, 214, 224, 303, 309.
Response 1: The format of this section has been reformatted so that the index of each reference is now correctly linked to its reference.
Point 2: Please zoom in on Figures 1 and 3 for a better visualization of the results.
Response 2: The dimensions of Fig. 1 and 3 have been enlarged for a better visualization of the results.
Point 3: Since A. thaliana is widely used in the fields of plant science, genetics and evolution and has helped further our understanding of germination and aspects of plant growth that are important in commercial crops, I recommend future research in this area.
Response 3: Thank you for your suggestion and we will consider further research in Arabidopsis.